# Development of Prediction Capabilities for High-Throughput Screening of Physiochemical Properties by Biomimetic Chromatography

**DOI:** 10.3390/molecules30234528

**Published:** 2025-11-24

**Authors:** Damian Tuz, Damian Smuga, Tomasz Pawiński

**Affiliations:** 1Laboratory of Physicochemical Analysis, Department of Medicinal Chemistry, Celon Pharma S.A., Marymoncka 15, 05-152 Kazuń Nowy, Poland; damian.smuga@celonpharma.com; 2Department of Drug Chemistry, Pharmaceutical and Biomedical Analysis, Faculty of Pharmacy, Medical University of Warsaw, Banacha 1, 02-097 Warsaw, Poland; tomasz.pawinski@wum.edu.pl

**Keywords:** high-throughput, screening, biomimetic chromatography, machine learning

## Abstract

The ever-increasing costs of in vitro and in vivo testing are compelling scientists to increasingly rely on computational models for predictive characterisation at early stages of drug discovery and development. The complexity of this stage requires high-throughput screening methods that can rapidly generate comprehensive information about new chemical compounds. This review explores innovative approaches assessing pharmacokinetic and pharmacodynamic properties of new chemical entities, with a focus on integrating machine learning as a transformative analytical tool. Machine learning algorithms are highlighted for their capability to train sufficient predictors combining biomimetic chromatography data (a high-throughput alternative for several physicochemical assays) with molecular features and/or molecular fingerprints obtained in silico and in vivo data of known compounds to allow efficient prediction of in vivo data for new chemical entities. By synthesising recent methodological advancements and giving useful practical approaches, the review provides insights into computational strategies that can significantly accelerate compound library screening and drug development processes.

## 1. Introduction

A significant focus in drug discovery involves establishing pharmacokinetic and pharmacodynamic properties through computational models. Pharmacokinetics describes the disposition of a drug in the body, encompassing the processes of Absorption, Distribution, Metabolism, Excretion, and Toxicity (ADMET), often summarised as “what the body does to the drug”. Meanwhile, pharmacodynamics pertains to the drug’s mechanism of action and its biochemical and physiological effects on the body, which can be summarised as “what the drug does to the body”.

The relationship between a molecule’s structure and its biological activity is governed by its fundamental physicochemical properties. Among these, lipophilicity—the affinity of a molecule for a lipid-like environment—is of paramount importance in Medicinal Chemistry. This property influences a compound’s entire ADMET profile, affecting its absorption, its distribution across biological membranes into various compartments in the body, its tendency to bind to plasma proteins, and even its potential for toxicity (Table 1).

To quantify lipophilicity, the logarithm of the n-octanol–water partition coefficient (LogP) is used, as it is the most common parameter for predicting physicochemical properties [2,3]. This value is determined using a reference system of two immiscible liquids, n-octanol and water, where n-octanol acts as a surrogate for biological lipid membranes. A compound’s LogP value measures its distribution equilibrium between these two phases, with higher values indicating greater lipid solubility (lipophilicity) and lower values signifying greater water solubility (hydrophilicity). Since LogP refers to neutral species, it is easy to compare values among neutral compounds, but it may be misleading for ionisable compounds. Because of this, there is another distribution coefficient, logD, which accounts for pH [4].

The shake-flask method is widely regarded as the gold standard for experimentally determining LogP/logD. This method involves partitioning a compound between immiscible solvents, octanol and water, and measuring the equilibrium concentrations in each phase. While highly accurate, the shake-flask method is time-consuming, requires high-purity compounds, and cannot be used for unstable compounds or those with extreme lipophilicity, due to analytical detection limits [2,5,6].

To overcome these limitations, chromatographic techniques were developed. Reversed-phase (RP)-HPLC is one of the most common alternatives. This method relies on calibration plots based on compounds with a known Chromatographic Hydrophobicity Index (CHI) [7]. The CHI value itself estimates the percentage of organic solvent (e.g., acetonitrile) needed to elute the compound. This CHI value (0–100) can be mapped onto the traditional octanol–water logD scale using a linear equation to produce ChromlogD. ChromlogD is now widely used as a high-throughput alternative to the shake-flask method for assessing lipophilicity.

While ChromlogD addresses the throughput limitation, n-octanol and C18 stationary phases do not perfectly replicate the complexity of biological systems. To experimentally assess critical physicochemical properties (lipophilicity, permeability, protein binding) in a more biologically relevant manner, biomimetic chromatography (BC) has emerged as a reliable high-throughput technique, which has been proven to be superior in lipophilicity assessment within certain groups of chemical compounds [8,9,10,11].

Predictive models in pharmacokinetics link experimentally obtained data (A), such as retention factors from BC, with in silico derived molecular descriptors (B) or chemical fingerprints (C). The aim is to reliably forecast data that are usually resource-intensive and challenging to acquire (D), like in vivo efficacy [12], plasma protein binding (PPB) [13,14], or blood–brain barrier permeability (log BB) [15,16,17,18]. This approach is known as the Quantitative Structure–Retention Relationship (QSRR) [19] (Figure 1).

The translation of raw chromatographic and in silico data into predictions of complex biological phenomena is greatly improved by using ML. This review will therefore examine these methods as follows:
Section 2 will explore the quantitative relationship between retention behaviour in these BC systems and their corresponding “gold standard” biological assays.Section 3 will introduce modern ML algorithms as tools to decode these complex, non-linear relationships and their use in QSSR.Section 4 will present selected applications from the recent literature, followed by our conclusions on the current state of the art and upcoming challenges.

## 2. Physiochemistry in Pharmacokinetics: From Gold Standards to Biomimetic Alternatives

The main physicochemical parameters are lipophilicity, permeability, and protein binding. However, they should not be viewed as independent variables. Collectively, these parameters offer essential information for understanding and predicting the behaviour of drug candidates in biological systems. While these parameters can be measured individually in vitro, their real value lies in forecasting outcomes in complex in vivo studies or cell-based assays (“golden standards”), which are resource-intensive and low-throughput.

This chapter examines how BC offers high-throughput alternatives for predicting these in vivo parameters. BC is an analytical method within (ultra) high-performance liquid chromatography ((U)HPLC). It may use artificial stationary phases designed to mimic the molecular interactions between pharmaceutical compounds and their biological targets, including proteins, cellular membranes, and enzymatic systems [20]. Retention times from BC can be used to model parameters such as lipophilicity, protein binding affinity, and membrane permeability characteristics, which can then be utilised to model more complex parameters like human oral absorption (%HOA) or log BB.

Implementing BC provides benefits in drug development, as it offers a cost-effective alternative to traditional in vivo studies and aligns with High-Throughput Screening (HTS) methods (Table 2).

### 2.1. Plasma Protein Binding (PPB) and Volume of Distribution (V_D_)

According to the free drug hypothesis, only the unbound fraction of a drug is biologically active, as it can diffuse across cell membranes to interact with its target site [21]. Intensive PPB can lower the concentration of unbound drug, reduce its V_D_, decrease Cl, and necessitate higher doses to attain the desired effect, which presents a major risk of toxicity. Although, in some cases, it can be beneficial by prolonging the active effect of a drug through slow release from plasma proteins [22,23,24]. The affinity of a drug for common plasma proteins, such as α1–acid glycoprotein (AGP) and human serum albumin (HSA), provides valuable insight into its pharmacokinetic behaviour, influencing its V_D_, half-life (t_1/2_) and clearance (Cl) [25]. The V_D_, t_1/2_, and Cl are also dependent on membrane permeability, which is heavily influenced by lipophilicity.

The gold standard for measuring PPB is Equilibrium Dialysis (ED). This method physically separates a protein-containing solution from a buffer using a semipermeable membrane, allowing the free drug to reach equilibrium across it [26]. It is accurate but slow and not suitable for HTS. Regarding V_D_, t_1/2_, and Cl, these parameters are not obtained through a single assay. Their gold standard determination is an in vivo pharmacokinetic (PK) study. In such studies, drug concentration in plasma is measured over time, and V_D_, t_1/2_, and Cl are calculated from this curve using pharmacokinetic models.

Using BC with AGP and HSA columns has become a reliable HTS method for studying PPB [27]. In this technique, the retention time of a drug on protein-coated columns is measured to determine retention factors, log k_w(HSA)_ and log k_w(IAM)_, which are correlated with the drug’s binding affinity to plasma proteins. Several studies have demonstrated a strong correlation between these retention factors and in vivo data obtained in PPB studies [8,13,14,18,28,29,30,31,32]. It is important to note that in some of this work, researchers combine multiple BC columns for HTS of various aspects of drug disposition in the Central Nervous System (CNS) (log BB, fraction unbound in brain, unbound brain volume of distribution) [18].

Technical Details: AGP and HSA are both protein-based columns within Affinity chromatography, initially designed for chiral separation by exploiting the stereospecific binding pockets of the immobilised proteins. However, these columns have found additional applications in ADMET profiling and can be employed as tools for drug distribution and drug–drug interactions. AGP contains α1-acid glycoprotein, a major plasma protein that binds basic and neutral drugs. HSA contains immobilised human serum albumin, a key plasma protein that binds numerous drugs in the bloodstream [33]. Daicel Corporation offers a wide range of protein-based chiral selectors. HSA and AGP are available under the trade names CHIRALPAK HSA and CHIRALPAK AGP. They also supply columns with immobilised Cellobiohydrolase (CBH) and serum albumins from various animal species.

An alternative BC approach to predict PPB utilises Micellar Liquid Chromatography (MLC) [34,35,36]. MLC was also utilised for VD, t_1/2_, and Cl [34,35,37]. This technique operates in reversed-phase liquid chromatography (RPLC) mode with a mobile phase containing surfactant at a concentration above its critical micellar concentration (CMC). The most common stationary phases are C8, C18, and cyanopropyl, while the typical surfactants include the anionic sodium dodecyl sulphate (SDS), cationic cetyltrimethylammonium bromide (CTAB), and non-ionic Brij-35 [38].

Technical Details: The retention and separation process in MLC relies on a double equilibrium. This determines how the analyte distributes itself among three different microenvironments: (i) between the bulk aqueous mobile phase and the surfactant-coated stationary phase; and (ii) between the bulk aqueous mobile phase and the micellar aggregates in the mobile phase. Analytes that bind strongly to the micelles are slowed down compared to those in the aqueous phase [39]. Due to this mechanism, the retention factor log k_w(MLC)_ is proportional to the compound’s partitioning into lipids on the surface of a stationary phase and micelles, providing results that directly measure membrane affinity.

### 2.2. Oral Bioavailability (F), Human Oral Absorption (%HOA), Membrane Permeability

Oral bioavailability (F) is a crucial in vivo parameter that reflects the proportion of an administered dose reaching systemic circulation unchanged. It is a composite parameter influenced by %HOA (which includes aqueous solubility, intestinal permeability, and chemical degradation in the gastrointestinal (GI) tract) and first-pass metabolism [40,41]. It is important to distinguish between different types of permeability: (i) membrane permeability affecting drug absorption, which can be further defined as intestinal permeability and epithelium permeability; and (ii) BBB permeability, influencing drug distribution specifically through the BBB, which is essential for drugs targeting the CNS. A compound’s ability to cross the BBB is a critical and distinct parameter, essential for designing CNS-active drugs or, conversely, for excluding drugs from the brain to prevent neurotoxicity.

The gold standard for oral bioavailability is an in vivo PK study comparing the area under the curve (AUC) of plasma concentration following an oral dose to that of an intravenous (IV) dose. Aqueous solubility and chemical degradation are relatively easier to assess than membrane permeability. For membrane permeability, the gold standard is the Caco-2 and Madin-Darby canine kidney (MDCK) cell-based in vitro assay. They are slightly different; the Caco-2 assay uses human colon carcinoma cells and requires long differentiation, whereas the MCDK assay provides a faster alternative and is frequently used in transfected MDR1-MDCK variants to investigate P-glycoprotein efflux transport specifically [42,43,44]. An alternative, well-established approach to measuring passive membrane permeability is the Parallel Artificial Membrane Permeability Assay (PAMPA) [45]. The membrane on which PAMPA methods are based is artificial, can only measure passive diffusion, and can be “customised”, depending on the membrane the assay mimics [46,47,48]. The gold standard for BBB permeability is derived from in vivo studies in animals. It is expressed as the logarithmic ratio of the drug’s concentration in the brain to its concentration in plasma, resulting in the log BB value. A less commonly used descriptor of BBB permeability is logPS, which represents the permeability surface area product. The difference between them is that log BB measures concentrations at equilibrium, and log PS measures the initial permeability rate. This process is low-throughput and resource-intensive [49].

The primary BC alternative for predicting membrane permeability is Immobilised Artificial Membrane (IAM) chromatography [50,51,52,53,54]. The retention factor log k_w(IAM)_ is proportional to the compound’s partitioning into the phospholipid phase, providing a direct measure of membrane affinity. This method offers a high-throughput prediction of passive diffusion. IAM permeability has a medium correlation with Caco-2, but only after the inclusion of the molecular mass variable and the exclusion of compounds that undergo active transport [55]. However, in a different dataset, a weak correlation between IAM permeability and Caco-2 has been reported [36]. A medium correlation can also be observed with MDCK, after the inclusion of a parameter representing electrostatic interactions [56]. Both correlations are only valid for drug compounds that undergo passive transport. IAM cannot predict the effect of active transporters, just like PAMPA, a key limitation compared to cell-based assays [43]. Because oral bioavailability (F) is a composite property, log k_w(IAM)_ can potentially be part of a prediction model (e.g., predicting %HOA, due to the heavy influence of lipophilicity on solubility and permeability) [29,56]. IAM chromatography has also been successfully applied in studies on epithelial permeability [57] and for predicting log BB [15,16,17,18]. The phospholipid stationary phase serves as a model for the membranes of brain endothelial cells. Various IAM column variants share similar capabilities in predicting BBB permeability [16]. Measurement of epithelium, membrane (intestine), and BBB permeability can be performed using a single type of column, commercially available IAM.PC.DD2. The differences lie in (i) the pH of the buffer—usually pH 5.5–7.4 for skin permeability, pH 6.5–7.4 for intestinal permeability, and pH 7.4 for BBB permeability; and (ii) different molecular descriptors.

Technical Details: IAM columns are phospholipid-based columns. The first commercially available column was IAM.PC (phosphatidylcholine). IAM.PC.DD2 is the latest version [57]. A switch from “Type A” silica to “Type B” silica after 2018 caused significant differences in retention for acidic and basic compounds. Valko et al. [58] emphasises using new CHI(IAM) values for calibration of new columns, for better in vivo correlations. IAM columns can be further specialised by using other phospholipids as head groups; IAM.PE (phosphatidylethanolamine) shows differences in abundance in vivo [59] and IAM.SPH (sphingomyelin) can give unique insights into drug–neuron activity due to its rich presence in animal nerve tissue compared to phosphatidylcholine [60].

MLC can expand possible information by predicting epithelium permeability [61,62,63]. MLC can also be utilised as a tool for predicting passive membrane permeability, and what follows %HOA [36,64,65,66]. MLC permeability shows a moderate to strong correlation with the PAMPA [36,66] assay and a medium to good correlation with Caco-2, but only for selected compounds that permeate passively [36,64]. MLC is also employed to forecast BB permeability [15,67,68]. Most of the presented studies utilise C18 as the stationary phase and Brij-15 as the surfactant. Similarly, like with IAM, the key factor that distinguishes different permeability measurements is pH and molecular descriptors.

While the methods mentioned earlier (IAM, MLC) are HTS tools for modelling general membrane partitioning and passive permeability, Cell Membrane Chromatography (CMC) represents a different, more biologically complex approach that may be employed at a later stage of drug discovery. CMC is designed to examine specific drug–receptor interactions using biologically active membranes. This method excels in studying drug–membrane interactions [69,70,71].

Technical Details: The core of the CMC stationary phase consists of adsorbed (on activated silica gel) cell membranes, which were historically sourced from tissue cells (e.g., rabbit red and white cells, rabbit cardiomyocytes and rat vascular endothelial cells [72,73,74,75]), and now from high-expression recombinant cell lines with specific receptors (e.g., Vascular Endothelial Growth Factor Receptor 2 (VEGFR-2), Fibroblast Growth Factor Receptor-1 (FGRF1) [76,77]). This approach preserves the biological structure and activity of receptors, allowing for accurate simulation of in vivo interactions [78,79]. The adsorption of high-expression cell lines significantly enhances the sensitivity and accuracy of the method. The mechanism of retention is based on the specific recognition between the analyte and the membrane receptor. Ligands, such as drugs, selectively interact with membrane receptors adsorbed on silica gel, achieving chromatographic separation. A key parameter that can be measured is the equilibrium dissociation constant (K_D_), which reflects the affinity strength between a drug and its receptor. Methods such as frontal analysis and zonal elution are used within the CMC framework to calculate these K_D_ values. Although CMC is widely used, due to a lack of commercial availability, its usage in HTS is heavily limited. Column life is relatively short due to membrane receptors falling off the silica gel, thereby losing stability and reproducibility. Moreover, the amount of attached membrane receptors in CMC should be controlled for the accuracy improvement [80].

### 2.3. Toxicity (DIPL and hERG)

Toxicity is often linked to extreme values of physicochemical properties. One specific form, DIPL, is a condition where drugs bind excessively to phospholipids, leading to their accumulation within cell lysosomes [81,82]. Another specific form of toxicity is inhibition of hERG cardiac potassium channel, which is an important antitarget in early drug discovery [83].

For DIPL and hERG, the gold standard involves an in vitro cell-based assay (e.g., using fluorescent dyes) or in vivo histopathology. All these methods are extremely slow and resource-intensive.

IAM retention factor log k_w(IAM)_ data may be used to predict toxicity. While moderate phospholipid affinity is required for permeability, an extreme affinity is a well-established indicator of DIPL risk [12,84,85,86]. Other toxicities, like hERG channel binding, have been correlated with high general lipophilicity (ChromlogD). However, more precise approaches are shown in the literature that leverage both retention factors of k_w(IAM)_ and k_w(AGP)_ [87].

As toxicity relates usually to specific drug–receptor interactions, CMC could potentially excel in this field with a wide variety of papers related to the screening of complex samples and identification of harmful components [71,88,89,90,91,92,93,94,95,96,97].

### 2.4. General Technical Considerations

Excluding CMC and MLC, which have their own unique requirements for mobile phases, mobile phases in BC, especially for IAM, AGP, and HSA, are designed to resemble physiological conditions closely, and they are typically composed of a buffered solution and organic modifiers. Due to the narrow range of optimal pH for these columns (5.0–7.4), the most popular buffers are phosphate and acetate [7,11,27]. While it is possible to conduct experiments using purely aqueous mobile phases, this approach leads to significantly prolonged retention times for analytes. Such time-consuming analyses are fundamentally incompatible with the HTS ideology that dominates modern drug discovery, which demands rapid and efficient methods. The introduction of fast gradient methods by Valko et al. was a pivotal development that gave a chance to replace slow isocratic runs and enabled the rapid analysis required for HTS [11,27]. For IAM chromatography, acetonitrile is used almost exclusively as an organic modifier. However, technical optimisation for IAM columns shows that predictive capabilities are similar using methanol [98]. Literature indicates the possibility of using mass spectrometry detection to increase the throughput, enabling the “pooling” of multiple compounds in a single ejection. Russo et al. clearly indicate that this methodology is faster, more environmentally friendly and results from MS-friendly chromatographic conditions, having good correlation with results from standard phosphate buffers [99].

## 3. Machine Learning (ML): Translating Chromatographic Data into Predictions

In the previous chapters, we established BC as a high-throughput method for generating experimental data (e.g., log k_w(IAM)_, log k_w(HSA)_) that mimics specific biological interactions. However, the raw retention factor is not, by itself, a prediction. ML, especially supervised learning, provides a set of computational tools that can develop a model to connect this experimental data (A) with in silico descriptors (B/C) to predict the complex in vivo parameters (D) we ultimately focus on, such as log BB or PPB.

ML can be broadly divided into supervised learning (where labelled data means the output is known) and unsupervised learning (where unlabelled data means there is no output). These methods are often combined to build a final model or make inferences (Figure 2). Even simple regression can be implemented manually, but ML frameworks benefit from scalability and efficiency through the ability to chain different algorithms in pipelines (excluding deployment in this context) [100]. Moreover, in the presented applications, no component automatically acquires new data, preprocesses it, builds new iterations of predictions, and immediately deploys them into production, simply because of the limited amount of available data.

While complex ML techniques, such as deep neural networks, can capture intricate patterns, they often require substantial computational resources, extensive knowledge, and large datasets, and suffer from reduced interpretability. A reasonable approach, and one commonly seen in the literature, is to begin with simple, interpretable models like linear regression and progress to advanced techniques only when necessary [101].

The development and training of ML models follow a standard workflow:
Data acquisition—represents curation of comprehensive, representative datasets. The quality and integrity of training data directly influence model performance and can significantly compromise the model’s predictive capabilities and generalisation ability.Data preprocessing—transforms data, including feature scaling, normalisation and handling missing values. Many ML algorithms exhibit sensitivity to feature scale disparities, where features differing by orders of magnitude can disproportionately influence model training. Standard techniques include standardisation, min-max scaling, and log transforming for heavily skewed distributions.Data partitioning—involves division of the dataset into training (typically 60–80%), validation (10–20%) and test (10–20%) sets. Data partitioning can be performed randomly, which is most suitable for large, diverse, and evenly distributed datasets, or through rational splitting, such as scaffold splitting that divides by groups of molecules (with similar chemical scaffolds), thereby ensuring better model generalisation and reducing overfitting [102]. The exact proportion may vary based on dataset size and specific application requirements [103].Model training—an optimisation process where a loss function (e.g., Mean Squared Error (MSE), Cross-Entropy) quantifies the error between the model’s prediction and the actual values. This function acts as a performance indicator that models seek to minimise iteratively throughout the training process. The choice of loss function influences model behaviour, particularly in handling outliers.Model evaluation—assessment of the final model’s performance on the unseen test set using evaluation metrics (e.g., R^2^, Q^2^). Evaluation ensures an unbiased assessment of the model’s ability to generalise to new, unseen data points. If the data partitioning step is omitted, this step should include validation methods (e.g., Leave-one-out Cross-Validation, LOOCV). In specific frameworks, such as Quantitative Structure–Activity Relationship (QSAR)/QSRR, and in general scientific papers, statistical tests should also be evaluated (e.g., Fisher test, *t*-test) to ensure model significance.

The selection of both loss functions and evaluation metrics is dependent on the specific type of ML task being addressed. While the selection of loss functions depends on the model’s requirements, evaluation metrics can be chosen purely based on their interpretability and relevance to business or research objectives.

Although retention factors derived from BC provide valuable information, they often require additional variables to accurately model complex biological systems, such as log BB. However, when combining chromatographic retention data with in silico computed parameters characterising chemical structures, the QSRR term should be utilised.

### 3.1. Molecular Representations

Before a quantitative model can be built, a chemical structure must be converted into a computer-readable format, known as a molecular representation. This representation can capture the molecule’s identity at different levels, such as one-dimensional (1D) string (e.g., SMILES), 2D graphs (which define atomic connectivity), or 3D conformations (which define atomic coordinates). From these foundational representations, numerical features are generated for the ML model [104]. The numerical features derived from molecular representation generally fall into two broad categories.

The first, molecular descriptors are numerical values that capture diverse, interpretable properties (e.g., logP, pKa, Polar Surface Area (PSA)) that can be broadly categorised into topological, geometrical, electrostatic, quantum, physicochemical and pharmacophoric types. Molecular descriptors can greatly complement experimental data obtained from BC and help with predictive modelling and property calculations. However, despite this applicability, model performance heavily relies on the quality of those molecular descriptors [105,106] (Table 3).

Good practice in working with molecular descriptors involves [107]

Cleaning—handling missing values and descriptors with no variance.Normalising/standardisation—some ML algorithms like Support Vector Machine (SVM) and Artificial Neural Network (ANN) require features to be on the same scale.Feature selection/dimensionality reduction—removing low-variance descriptors, eliminating highly correlated descriptors, usually by unsupervised learning.Handling categorical variables—presence/absence of functional group requires encoding to binary (0, 1).Handling outliers—deciding if outliers should be included in the model.

The second, molecular fingerprints, encodes structural patterns such as binary vectors (bits) or bitstrings. Unlike molecular descriptors, molecular fingerprints are used without normalisation or standardisation. They encode narrow structural patterns in the form of bits that are not interpretable and can be categorised into substructural, topological, crystallographic, and hybrid types. They are best suited for similar searching and in silico screenings.

There are several tools and libraries, both commercial and open-source, for description calculation, like RDKit, Open Babel, Scikit-learn, or the KNIME 5.3 analytics platform [108,109,110,111]. These open-source cheminformatics tools are essential for calculating molecular descriptors and fingerprints used in the QSRR modelling. Scikit-learn is a comprehensive Python 3.14.0 library that provides tools for feature selection, data scaling, and model validation. KNIME is a perfect solution for researchers who do not want to learn a programming language.

### 3.2. Unsupervised Learning

Unsupervised learning aims to discover patterns, relationships, and exceptions within variables, through algorithms based on clustering (structure discovery and/or anomaly detection) and dimensionality reduction, which are often employed synergistically to extract maximum information. However, while anomaly detection algorithms are a great tool, for the datasets depicted in this work, easier statistical outlier detection is more appropriate and robust.

Clustering methodologies, particularly *k*-means clustering and hierarchical clustering, have demonstrated significant utility in cheminformatics [112,113]. In the topic of ML, dimensions are described by variables. Dimensionality reduction techniques aim to decrease the number of variables (dimensions). A Principal Component Analysis (PCA), first introduced by Pearson and later developed by Hotelling, identifies orthogonal components that maximise variance in multivariate data [114,115]. The PCA in QSRR can be used to reduce the number of molecular features generated in silico, thereby preprocessing data before applying supervised learning techniques. However, it is essential to note that PCA does not select but creates new principal components (variables) that retain information from potentially correlated variables. An identification and interpretation of outliers can indicate experimental artefacts or reveal compounds with unique binding mechanisms. Two prominent approaches for anomaly detection include Isolation Forest and Density Spatial Clustering of Applications with Noise (DBSCAN) [116,117]. Isolation Forest identifies anomalies through recursive partitioning of the feature space, while DBSCAN designates points in low-density regions as outliers based on spatial density distributions.

Evaluation metrics and loss functions serve different purposes depending on the specific unsupervised learning task. For instance, in clustering, the focus is often on measuring both the cohesion within clusters and the separation between clusters. In contrast, dimensionality reduction metrics typically focus on information preservation and the quality of reconstruction (Table 4).

Pastewska et al. [9,10] used hierarchical cluster analysis (CA) and PCA to explore relationships among different lipophilicity measures (both in silico and experimental, including IAM retention). They then applied the Sum of Ranking Differences (SRD), an evaluation metric used to systematically compare and rank those lipophilicity measures, determining which methods most reliably capture lipophilicity. Jeličić et al. [8] used hierarchical CA and PCA to evaluate similarities between calculated and experimentally observed values (including PPB measure from HSA and AGP, and lipophilicity from IAM) and their mutual correlation.

### 3.3. Supervised Learning

In supervised learning, the model learns a mapping function from labelled data. This is the most common approach for building predictive QSRR and pharmacokinetic models [123].

#### 3.3.1. Regression Models

Regression models are used to predict continuous outputs (log BB, log k_w(IAM)_). Regression models can be univariate, featuring only one variable, or multivariate, which depends on multiple variables. Linear regression assumes a proportional relationship between variables and the output. Their primary advantage is simplicity and high interpretability, because the resulting equation clearly shows which features are most important.

Linear regression assumes additive and proportional effects between any number of variables (X) and output (Y). When a non-linear relationship exists between variables and output, non-linear regression can capture curvilinear, saturating, or sigmoidal relationships commonly seen in biological systems. In linear regression, which predicts continuous outcomes, loss functions assess the difference between predicted values (ŷ) and actual values (y). Linear regression models are most commonly used in QSRR because of their simplicity and ease of interpretation (Table 5 and Table 6).

Linear regression is the most prevalent model in this field for these exact reasons. For example, De Vrieze et al. [15] created a multivariate linear regression (PLS model, LOOCV) to predict log BB using log k_IAM_ and log k_MLC_. In this work, they initially obtained models that properly fit the training data but poorly fit the validation set, indicating overfitting (average difference between R(PLS) and R(LOOCV) = 0.2407). After manually reducing the variables from 15 to 7, the difference dropped to 0.0928, indicating that models after dimensionality reduction enable more robust predictions, a common observation among prediction models. Later, De Vrieze et al. [16] dedicated themselves to evaluating the prediction power of log BB from IAM.PC.DD2, IAM-sphingo, and IAM-cholesterol, with a similar performing model, proved that each of these columns predicts log BB well. Janicka et al. [17] developed a regression model using Multiple Linear Regression (MLR) with identified key descriptors: log K_w(IAM)_ and molecular weight (MW). Statistical validation (R^2^ = 0.934, cross-validated Q^2^ = 0.934) confirmed the model’s robustness, with PCA ensuring descriptor independence. In more broad study, Vallianatou et al. [18] incorporated unsupervised learning, PCA and hierarchical CA, to obtain a clear overview of the dataset, based on which they applied two (MLR and Partial Least Squares (PLS)) models to create an HTS approach for early drug evaluation of CNS drugs, modelling not only log BB, but also unbound fraction in brain and unbound brain volume of distribution.

Due to the high complexity of biological prediction, a linear relationship can be insufficient. More complex non-linear algorithms like Support Vector Regression (SVR), Random Forest Regression (RFR), and ANN can capture curvilinear, saturating, or sigmoidal relationships [124].

A clear example of using non-linear regression is the work of Tsopelas et al. [56] focused on modelling %HOA, which captures a sigmoidal relationship with retention factors obtained from IAM chromatography and several molecular descriptors. The study of Ciura et al. [125] compared MLR with non-linear ANNs, using both multilayer perceptron (MLP) and radial basis function (RBF) architectures. The models were constructed using 261 experimental CHI_IAM_ values combined with four molecular descriptors: PSA, MV, HDo, and distribution coefficient at pH 7.4 (logD 7.4) to predict CHI_IAM_. While the linear MLR model showed moderate predictive ability (R^2^ = 0.550, Q^2^ = 0.513), the best MLP network (ANN4) demonstrated superior performance with higher R^2^ values for both training (0.746) and external validation (EV) (0.677) sets. Global sensitivity analysis identified PSA and MV as the most influential descriptors, emphasising the importance of polar interactions and molecular size in drug–membrane interactions.

**Table 5 molecules-30-04528-t005:** Summary of differences between linear and non-linear regression.

Aspect	Linear Regression	Non-Linear Regression
Interpretability	Directly interpretable coefficients	Parameters are often context-dependent
Flexibility	Limited to linear trends	Captures saturating, sigmoidal or exponential relationships
Models	Ordinary Least Squares (OLS), PLS, MLR, SVR [126]	Polynomial Regression, SVR [127], RFR [128], Extreme Gradient Boosting (XGBoost) [129]

**Table 6 molecules-30-04528-t006:** Set of three loss functions, validation methods, evaluation metrics, and statistical tests usually used in supervised learning (QSRR).

Loss Functions	Validation Method	Evaluation Metrics	Statistical Test
MSE—calculates the average of the squared differences between ($\hat{y}$) and (y). Heavily penalises significant errors and is sensitive to outliers. Assuming errors follow a Gaussian distribution.	LOOCV trains on *n*-1 samples, tests on 1, and repeats n times. It is ideal for small datasets. Offers an unbiased estimate but comes with a high computational cost.	Sum of Squared Errors (SSE)—measures the total squared error between predictions and actual values.	F-test—overall model significance. Checks if the prediction is not due to chance alone. Standard threshold: *p* < 0.05.
Mean Absolute Error (MAE)—calculates the average absolute differences between ($\hat{y}$) and (y). It is robust for outliers. Suitable for data containing outliers or errors that follow the Laplace distribution.	k-fold Cross-Validation (k-Fold CV). Splits data into k equal parts. Each fold is used only once as a test set.	R^2^ or R (e.g., “ML Model”)—measures the proportion of variance in the dependent variable that is explained by the independent variable. It ranges from 0 to 1, where R^2^ = 1 means that the model explains all the variance, and R^2^ = 0 implies that the model explains none of the variance.	*t*-test—individual variable significance. Provides *p*-value for each variable. Standard threshold: *p* < 0.05.
Huber Loss—combines MSE for minor errors (smooth and differentiable) and MAE for significant errors (robust to outliers) [130].	EV (train–test split). Splits data into training set and independent set. The test set should never be used to train models. Gold standard for a prediction model. Provides R^2^_ext.	Q^2^ or R (e.g., “ML Model” with “Validation method”)—used to evaluate the predictive performance of a model, particularly in cross-validation or EV scenarios. It measures how well the model predicts new, unseen data. It ranges from −∞ to 1, where Q^2^ = 1 means strong predictive power, and Q^2^ < 0 implies that the model performs poorly on unseen data [131,132].	Y-randomisation—permutation test. Checks robustness by verifying that R^2^ and Q^2^ remain similar after a random change to the output value. If R^2^ and Q^2^ drop in values, it is good because a relationship exists between the variables and the output [133].

#### 3.3.2. Classification Models

Classification models are supervised ML algorithms designed to predict discrete class labels (e.g., 1/0, active/inactive, toxic/non-toxic) based on input features. Unlike regression, which predicts continuous outcomes, classification assigns data points to predefined categories, making it helpful for decision-making in drug discovery [100] (Table 7 and Table 8). The type of classification model depends on the number of labels that need to be populated. In the case of binary labels, there are two popular models: logistic regression, which estimates class probabilities using the logistic function, and SVM, which finds a hyperplane that maximises the margin between classes [123]. In the case of nonbinary classification, such as classification into different drug families, there are two other classic models: multinomial logistic regression, which extends logistic regression to multiple classes, and RFR. In some cases, complex classification (binary or categorical labelling) may not represent the most accurate or informative approach. Instead, probabilistic outputs that quantify the likelihood of class membership often provide a superior strategy for capturing prediction confidence and variability [100]. Two probabilistic classification models are Naïve Bayes, which applies Bayes’ theorem with feature independence assumptions, and ANN, which uses SoftMax activation for probability distribution. Loss functions in classification models quantify the accuracy of a model’s predictions by measuring the discrepancy between the predicted probabilities and the actual labels. Those types of ML algorithms are not usually used to correlate with chromatographic data, but rather as a classifier combining different kinds of information; for example, in decision-making for drug discovery. Clever usage of this approach has been presented in the work of Tsopelas et al. [54], where, based on a trained non-linear regression model, they developed a simple classification approach.

## 4. Discussion and Future Perspectives

The application of biomimetic principles to pharmaceutical sciences has evolved significantly since the early works of Valko et al. [11,27]. Over the last two decades, the field has advanced from simple isocratic measurements to automated HTS platforms. However, a critical review of the recent literature reveals that while instrumental capabilities have expanded, the field often tends toward incremental improvements rather than transformative innovation.

### 4.1. Throughput vs. Mechanistic Understanding

A recurring challenge in BC is the trade-off between analytical speed and the depth of mechanistic understanding.

Bunally et al. [12] marked a significant shift by introducing a 96/384-well plate format, integrating multiple parameters (ChromlogD, HSA binding, membrane interaction) into a single automated workflow. This addressed the speed bottleneck but potentially simplified the biological interpretation.Russo et al. [50] demonstrated the viability of 2D-LC systems combining HSA and IAM columns. Their work on a visual clustering approach for permeability characterisation offered an alternative to traditional statistical modelling.Vallianatou et al. [18] proposed complex HTS approach for early-stage CNS drug candidates.Conversely, Iwakuma et al. [84] dived into the detailed mechanism of drug–membrane interactions in chromatographic separation on IAM stationary phase. Investigating acetonitrile concentrations and salt effects.Alternative approaches like those by Ciura et al. [144] using micellar electrokinetic chromatography (MEKC) raises fundamental questions about whether complex biomimetic surfaces are even necessary, as high correlations (R^2^ = 0.904) were achieved with simplified surfactant systems.

### 4.2. Analytical Bottleneck

Despite the development of fast gradient methods, a significant number of studies still rely on long gradients or isocratic methods to preserve column life or peak resolution. Furthermore, the field exhibits an over-reliance on ultraviolet (UV) detection, as shown by Russo et al. [99]. In response, they proposed a mass spectrometry-based approach, demonstrating that this methodology is faster, more environmentally friendly, and yields results that correlate well with those from standard phosphate buffers. However, it is another trade-off, because switching from phosphate-buffered saline (PBS) buffers to ammonium acetate in different datasets may result in reduced biomimicry [20].

### 4.3. From Regression to Black Boxes

It is important to remember that the performance of various ML models should be assessed using the same datasets. The number and distribution of samples have a significant influence on model performance. For instance, a small number of samples with similar structures may produce good predictions during cross-validation (high R^2^, high Q^2^). However, when tested against external datasets, its effectiveness often diminishes (R^2^_ext) due to differences in structure across the extensive chemical space. This is a typical sign of a model with a limited applicability domain.

The field clearly progresses from linear regression to complex pipelines. Use of unsupervised learning is much more common in more recent works, clearly benefiting researchers through graphical and statistical relationships between variables [9,10]. Ciura et al. [125] advanced this further by deploying an ANN, which showed superior performance to simple regression. While innovative, the use of increasingly complex ML algorithms threatens to obscure mechanistic understanding through the development of black-box models. There is a risk of prioritising predictive efficiency over understanding the underlying biological issues. However, the answer can be in the middle, like “gray box” ML algorithms [145].

This problem may be exacerbated by the rise and rapid improvement of large language models (LLMs). While they offer immense coding and analytical assistance, they may pose a trap for inexperienced analysts by luring them into black-box models that predict well but are incomprehensible to humans.

### 4.4. Will in Silico Replace Experimental?

With the increasing availability of curated chemical databases, data-intensive approaches like deep convolutional neural networks pose a risk of making experimental BC obsolete in the future. However, the current literature remains contradictory regarding the predictability of physicochemical parameters through descriptor-based models alone. As shown by the comparative studies of Orzel et al. [53] and Iwakuma et al. [86], experimental validation remains necessary to capture the dynamic nuances of biological environments.

## 5. Conclusions

This review highlights that BC has successfully matured from a niche analytical technique into a robust partner for ML in early drug discovery. By combining the high-throughput generation of biologically relevant data (IAM, HSA, AGP, MLC) with advanced QSRR modelling, researchers can now estimate in vivo parameters, such as log BB, %HOA, and specific toxicity, with increasing accuracy.

## Figures and Tables

**Figure 1 molecules-30-04528-f001:**
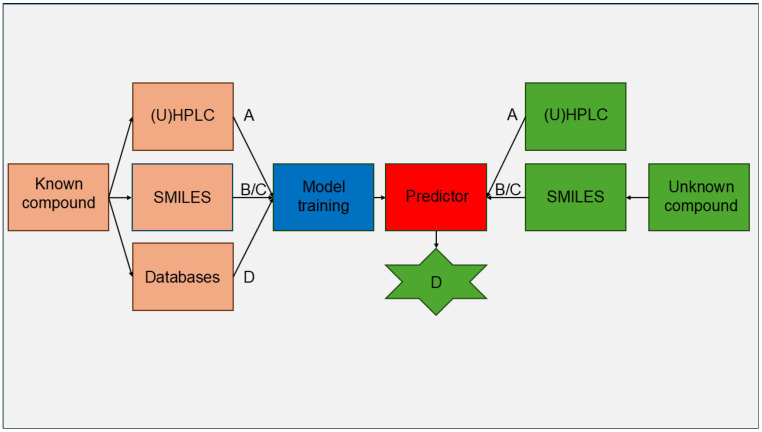
Flowchart of a prediction model trained with compounds that have publicly accessible in vivo data used to forecast the relevant in vivo data for unknown compounds. A—experimentally derived data, e.g., retention time; B—molecular descriptors generated in silico; C—molecular fingerprints generated in silico; D—in vivo data.

**Figure 2 molecules-30-04528-f002:**
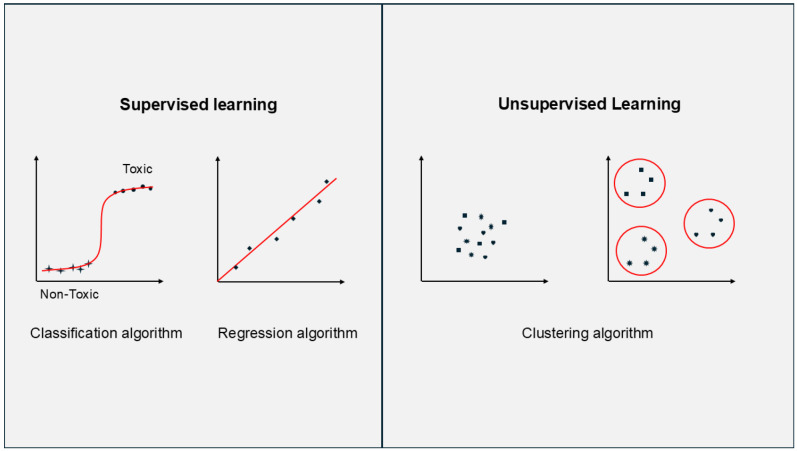
Simple graphical representation of difference between supervised learning (using labelled data) and unsupervised learning (using unlabelled data).

**Table 1 molecules-30-04528-t001:** Connection between lipophilicity and ADMET [1].

Drug	Parameters Influenced by Lipophilicity
Absorption	Solubility Membrane Permeability
Distribution	Blood–Brain Barrier (BBB) permeability Volume of distribution (V_D_)
Metabolism and Excretion	Susceptibility to oxidative metabolism Half-life (t_1/2_) Clearance (Cl)
Toxicity	Drug-induced phospholipidosis (DIPL) human ether-a-go-go-related gene (hERG) toxicity

**Table 2 molecules-30-04528-t002:** Comparative table for BC techniques.

Method	Stationary Phase (Characteristics)	Application Area	Advantages	Disadvantage
IAM	Phospholipids (e.g., phosphatidylcholine) are covalently bonded to silica	Membrane permeability, BBB permeability, lipophilicity (phospholipid affinity), phospholipidosis risk, human oral absorption, protein binding	Commercially available (Regis), robust, and HTS-compatible, this is a suitable model for passive diffusion	Does not model active transport. “Type B” silica switch caused retention shifts.
Affinity (HSA/AGP)	Immobilised plasma proteins on a support	Plasma protein binding, chiral separations	Commercially available (Daicel), HTS-compatible, directly measures binding to key plasma proteins	Specific protein binding. Does not measure membrane permeability or lipophilicity.
CMC	Immobilised, intact cell membranes or whole cells (e.g., HEK 293)	Drug–membrane interactions, specificity testing	Provides the most biologically relevant model	Not commercially available, complex to prepare, lower stability and robustness, HTS-incompatible.
MLC	Standard RPLC phase (e.g., C18) with a micelle-containing mobile phase (e.g., SDS, CTAB)	Membrane permeability, BBB permeability, lipophilicity, human oral absorption, protein binding	Uses standard columns, cost-effective, versatile (surfactant choice alters properties)	Complex separation mechanism (dual equilibrium). Micelles may not perfectly mimic biological membranes.

**Table 3 molecules-30-04528-t003:** Summary of differences between different molecular descriptors [106].

Type of Descriptor	Definition	Examples	Application
Topological	Derived from molecular graphs and encodes information about the connectivity and branching of atoms in a molecule	Degree of branching, Molecular connectivity indices, Wiener index	Solubility Boiling point Biological activity
Geometrical	Encode information about the 3D shape and size of the molecule	Molecular surface area (MSA), molecular volume (MV), principal moments of inertia	Molecular interactions
Electrostatic	Quantify the distribution of electric charge within a molecule	Partial atomic charges, dipole moment, Eeectrostatic potential maps	Hydrogen bonding Ionic interactions.
Quantum	Derived from quantum mechanical calculations	HOMO-LUMO gap, ionisation potential, electron affinity	Chemical reactivity Stability Spectroscopic properties
Physicochemical	Represent physical and chemical properties of molecules	logP, pKa, PSA	ADME properties
Pharmacophoric	Represent the spatial arrangement of features in a molecule that are essential for biological activity	Hydrogen bond donors (HDo), hydrogen bond acceptors (HAc), aromatic rings	ADME properties

**Table 4 molecules-30-04528-t004:** Description of the most popular loss function and evaluation metric for different unsupervised ML algorithms.

Method Category	Clustering	Dimensionality Reduction	Anomaly Detection
Loss function	Silhouette Coefficient—evaluates cluster cohesion and separation by measuring how similar points are to their own cluster compared to other clusters. Ranges from −1 to 1, where higher values indicate better-defined clusters [118].	Kullback–Leibler Divergence—measures the difference between probability distributions of original and reduced-dimensional data. Widely used in variational autoencoders. Lower values indicate better preservation of data structure [119].	Isolation Score—Measures how easily a point can be isolated from the rest of the data through random partitioning. Lower values indicate a higher likelihood of being an outlier [116].
Evaluation metric	SRD—evaluate how different clustering algorithms rank or group similar objects [120]. Inertia measures the sum of the squared distances between each data point and its closest centroid, commonly used in k-means. Lower values indicate better-defined clusters [112].	Reconstruction error—Quantifies the difference between the original data and its reconstruction after dimensionality reduction, significant in autoencoders. Lower values indicate better preservation of information [121].	Local Outlier Factor (LOF) Score—Compares the local density of a point with the densities of its neighbours. Higher values indicate a more substantial likelihood of being an outlier [122].

**Table 7 molecules-30-04528-t007:** Summary of popular classification models.

Model	Logistic Regression [134]	Decision Trees [135,136]	SVM [137]	ANN [138,139,140]
Strengths	Interpretable, efficient with small data	Handles non-linear data, interpretable	Effective in high-dimensional spaces	Captures complex patterns, scalable
Limitations	Limited to linear decision	Prone to overfitting	Computationally intensive with large data	Require large datasets, poor interpretability
Use Case	Binary toxicity	Rule-based ADMET screening	Drug–target interaction prediction	Multi-task toxicity

**Table 8 molecules-30-04528-t008:** Description of three popular loss functions and five evaluation metrics for classification models.

Loss Functions for Classification Models	Evaluation Metrics for Classification Models
Cross-entropy loss (Log Loss)—measures the difference between predicted class probabilities and true labels [141].	Accuracy—ratio of total correct predictions (both positive and negative) out of all predictions. Best for balanced sets.
Hinge loss—used for margin maximisation in SVM. Penalises predictions that are on the wrong side of the decision boundary [142].	Precision—ratio of correctly predicted positive instances out of all the cases predicted as positive. Measure how reliable an optimistic prediction is.
Focal loss—addresses class imbalances by focusing on complex classifiable examples. Gives small weight to easy examples [143].	Specificity—precision, but for pessimistic predictions.
	Recall (sensitivity)—ratio of actual positive instances that the model correctly identifies.

## Data Availability

No new data were created or analyzed in this study. Data sharing is not applicable to this article.

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
