# Peer review of "Development of Prediction Capabilities for High-Throughput Screening of Physiochemical Properties by Biomimetic Chromatography"

_molecules, 2025, doi:10.3390/molecules30234528_

Round 1
Reviewer 1 Report
Comments and Suggestions for Authors
The manuscript is interesting and well-written. I have some minor comments:
Line 99: The acronym "ADMET" is not defined. "ADME" has been only defined. Please comment that the letter "T" stands for toxicity.
Line 107-108: "..HSA and AGP... designed to model specific biological interfaces and binding phenomena": In fact, the initial aim that these stationary phases were developed was to be utilized in chiral separations.
Lines 118-120: Some comments about the recent replacement of fine-pored ‘Type A’ silica (solid support f the column) by medium-pored ‘Type B’ silica of the commercially available IAM stationary phases by Regis would be helpful for the readers. Type B silica has similar physical
properties as type A although it is less acidic. This implies that differences may be observed for acidic and basic molecules.
Lines 138-142: Please mention and explain the double equilibrium existing in micellar chromatography (the primary between the bulk aqueous phase and the surfactant- coated stationary phase as well as be- tween aqueous phase and micellar aggregates).
Lines 143-148: I suggest authors to mention that in order to increase throughput of biomimetic chromatography, MS detector can be used (in order mixtures of 10, 20.. compounds) to be injected each time. In this case, mobile phase should be compatible with MS detector (phosphate buffers should be avoided). For example, ammonium acetate buffers are suggested. In this case, a loss of biomimicy may be observed (due to the replacement of the buffer) and a calibration may be appropriate.
Table 1: (a) BBB penetretion refers to distribution and not to absorption, (b) Toxicity is not closely related to solubility. On the contrary, very high logP may imply bioaccumulation and extensive metabolism and may be connected to hepatotoxicity.
Line 190: Please do not use the term "calibration curve". Use instead the term "calibration plot" as we always work in the linear part of the graph.
Lines 516-517: Abbreviations: Please add ADME (as ADMET without toxicity),
Author Response
Dear Reviewer,
We want to thank you for taking the time to review our paper. Some of your comments directed our attention to explore certain parts of the manuscript, deepening our understanding of the topic.
Before we address your comments line by line, we would like to point out that the manuscript has undergone significant revisions. The first reason for this was to incorporate the content of Table 8 into the core of the manuscript. The second reason was to enhance the depth and completeness of my work. Major changes include:
- Introduction: We relocated lipophilicity from the physicochemical section here to avoid repetition. This allowed us to establish a clear structure: physicochemical problem, golden standard, biomimetic chromatography solution.
- Chapter 2: We have combined biomimetic chromatography and physicochemical sections into one to strengthen their connection. This section also follows the structure outlined in the introduction. Additionally, we have expanded on two techniques that previously received only superficial treatment.
- Chapter 3: We decided to rearrange the subchapters to reflect better the sequence that an analyst should follow when building a prediction model. It begins by emphasising the retention data and structural features, then implements unsupervised learning, and concludes with supervised learning. We have also moved specific examples of these techniques to demonstrate better the connection between this chapter and the previous one.
- Chapter 4: We have integrated discussion and future perspectives to showcase the current state of the art and future challenges.
Regarding specific comments:
Line 99: The acronym „ADMET” is not defined. “ADME” has been only defined. Please comment that the letter “T” stands for toxicity.
- The acronym ADMET was introduced with Toxicity right at the start and has been used throughout the work, to avoid confusion – line 33.
Linea 107-108: “…HSA and AGP…” designed to model specific biological interfaces and binding phenomena”: In fact, the initial aim that these stationary phases were developed was to be utilized in chiral separations.
- The relevant line concerning the origin of chiral separation has been added – lines 140-142.
Lines 118-120: Some comments about the recent replacement of fine-pored “Type A” silica (solid support f the column) by medium-pored “Type B” silica of the commercially available IAM stationary phases by Regis would be helpful for the readers. Type B silica has similar physical properties as type A although it is less acidic. This implies that differences may be observed for acidic and basic molecules.
- The introduction to this issue has been added – lines 205-209.
Lines 138-142: Please mention and explain the double equilibrium existing in micellar chromatography (the primary between the bulk aqueous phase and the surfactant coated stationary phase as well as between aqueous phase and micellar aggregates
- The paragraph about MLC was greatly expanded, including an interesting mechanism of retention – lines 159-169.
Lines 143-148: I suggest authors to mention that in order to increase throughput of biomimetic chromatography, MS detector can be used (in order mixtures of 10, 20… compounds) to be injected each time. In this case, mobile phase should be compatible with MS detector (phosphate buffer should be avoided). For example, ammonium acetate buffers are suggested. In this case, a loss of biomimicry may be observed (due to replacement of the buffer) and a calibration may be appropriate.
- This comment was particularly important for us because initially, we thought that nobody had tried it before. However, upon further inspection, we found only one literature example that uses pooling and mass spectrometry as detection methods – line 292-296 and 547-553.
Table 1: (a) BBB penetration refers to distribution and not to absorption, (b) Toxicity is not closely related to solubility. On the contrary, very high logP may imply bioaccumulation and extensive metabolism and may be connected to hepatotoxicity.
- The table was remodelled to be less confusing with ellipsis. (a) BBB permeability was correctly separated from membrane permeability and moved to distribution; (b) We have removed this untrue association – Table 1.
Line 190: Please do not use the term “calibration curve”. Use instead the term “calibration plot” as we always work in the linear part of the graph.
- We have changed 'calibration curve' to 'calibration plot' – line 61.
Linea 516-517: Abbreviations: Please add ADME (as ADMET without toxicity).
- We have decided to remove ADME and replace it solely with the ADMET – Abbreviations section.
I hope this new version will be more enjoyable to read and make a more significant contribution to its field.
Regards,
Damian Tuz

Reviewer 2 Report
Comments and Suggestions for Authors
- This paper summarizes the fundamental methodologies and concepts related to high-throughput screening (HTS) techniques for testing physicochemical properties in drug discovery, along with a machine learning model for prediction. Additionally, it presents a comprehensive table that encapsulates all relevant research study results.
- This paper is well-narrated with a logical flow and is easy to understand; however, the information lacks sufficient depth and completeness to provide proper guidance to the community.
- Biomimetic chromatography section:
- This section discussed about the available high throughput techniques, including IAM, CMC and MLC. It would be better to provide graphics to illustrate the differences for different techniques. Alternatively, it could also provide a table to make comparisons for different methods, with the columns as method, application area, characteristics, advantages, and disadvantages.
- The authors place considerable emphasis on IAM and its derivatives. In my opinion, other methods should receive equal attention.
- Although I knew that the author provided table 8 to discuss how different biomimetic chromatography methods can be used to test the physiochemistry properties, it would be better to show how these methods correctly measure the potential drug properties, implying it has better drug-likeness. For example, the golden standard of testing cell permeability is to use Caco-2 or MDCK cell to test drug permeability, but it would be very expensive and time consuming. Therefore, alternatively, it could test LogP or ChromLogD to predict the permeability, because they have correlation. Then, it would better to show and discuss the correlation between Caco-2 results and LogP results, etc.
- Physiochemistry section:
- This section discusses physicochemical properties relevant to early drug discovery but does not cover all crucial aspects, such as oral bioavailability and toxicity. While it is understood that various physicochemical properties are interconnected, it would enhance the discussion to explain how these properties influence one another.
- In the discussion of lipophilicity, the authors mentioned about the LogP, but only provided one sentence discussing about chromLogD. It would better to discuss more about that.
- Machine learning or prediction models section:
- The author provided basic information of machine learning, but it did not link to the physiochemistry properties prediction and motivations of using machine learning model. Although I understand that table 8 provided a lot of information for the current available papers using machine learning or models to predict physiochemistry properties, I would suggest that when the authors discuss about the machine learning techniques, they could also provide the examples of existing models using those techniques for prediction, which might have disadvantages or advantages. The authors can discuss about the reason or suggestion why it would better to use that machine learning techniques.
- Discussion section:
- While Table 8 offers useful information, it is challenging for readers to grasp the messages the authors intend to convey to the community. The comments are quite general, and the second column contains excessive text, which may not be appropriate. I suggest making this section more concise and precise.
- Biomimetic chromatography section:
- Specific comments:
- Table 1: In the second column, it showed ellipsis, leading to a lot of confusion for readers. I would suggest to make the table into paragraph or make the second column only showing some key words, and then discuss that in the manuscript.
- In line 171, please provide more discussion about ChromLogD.
- In lines 215-217, the authors only mentioned about Caco-2 and PAMPA. It would better to also mention MDCK, and the difference between these assays. The authors also mentioned that the correlation is weak for compounds. Could you provide more detailed discussion about that? Because that would be very insightful information which the community would like to know.
- Table 1 and 7 contain ellipsis, which is very difficult to follow.
Author Response
Dear Reviewer,
We want to thank you for taking the time to review our paper. We greatly appreciate your critics regarding our paper.
Before we address your comments line by line, we would like to point out that the manuscript has undergone significant revisions. The first reason for this was to incorporate the content of Table 8 into the core of the manuscript. The second reason was to enhance the depth and completeness of my work. Major changes include:
- Introduction: We relocated lipophilicity from the physicochemical section here to avoid repetition. This allowed us to establish a clear structure: physicochemical problem, golden standard, biomimetic chromatography solution.
- Chapter 2: We have combined biomimetic chromatography and physicochemical sections into one to strengthen their connection. This section also follows the structure outlined in the introduction. Additionally, we have expanded on two techniques that previously received only superficial treatment.
- Chapter 3: We decided to rearrange the subchapters to reflect better the sequence that an analyst should follow when building a prediction model. It begins by emphasising the retention data and structural features, then implements unsupervised learning, and concludes with supervised learning. We have also moved specific examples of these techniques to demonstrate better the connection between this chapter and the previous one.
- Chapter 4: We have integrated discussion and future perspectives to showcase the current state of the art and future challenges.
Regarding specific comments:
Biomimetic chromatography section:
This section discussed about the available high-throughput techniques, including IAM, CMC and MLC. It would be better to provide graphics to illustrate the differences for different techniques. Alternatively, it could also provide a table to make comparisons for different methods, with the columns as method, application area, characteristics, advantages and disadvantages.
- Table 2 has been introduced. It compares the biomimetic chromatography columns in stationary phase, application areas, advantages, and disadvantages – Table 2.
The authors place considerable emphasis on IAM and its derivatives. In my opinion, other methods should receive equal attention.
- The new Chapter 2 should have a more balanced approach between biomimetic chromatography methods. We have expanded sections for MLC and CMC – Chapter 2.
Although I knew that the author provided table 8 to discuss how different biomimetic chromatography methods can be used to test the physiochemistry properties, it would be better to show how these methods correctly measure the potential drug properties, implying it has better drug-likeness. For example, the golden standard of testing cell permeability is to use Caco-2 or MDCK cell to test drug permeability, but it would be very expensive and time consuming. Therefore, alternatively, it could test LogP and ChromLogD to predict the permeability, because they have correlation. Then, it would better to show and discuss the correlation between Caco-2 results and LogP results, etc.
- The manuscript follows a clear structure: physicochemical problem, golden standard, biomimetic chromatography. This allows us to better present how these methods accurately measure the potential of drugs – Chapter 2.
Physiochemistry section:
This section discusses physicochemical properties relevant to early drug discovery but does not cover all crucial aspects, such as oral bioavailability and toxicity. While it is understood that various physicochemical properties are interconnected, it would enhance the discussion to explain how these properties influence one another.
- The physiochemistry chapter has been merged with biomimetic chromatography. We have revised the structure of the subsections to emphasise the in vivo parameters discussed in Chapter 2.
In the discussion of lipophilicity, the authors mentioned about the logP, but only provided one sentence discussing about chromlogD. It would better to discuss more about that.
- The previous issue was dividing attention to lipophilicity between the introduction and the physiochemical section. We have moved lipophilicity entirely to the Introduction as a key parameter influencing ADMET, with the structure guiding the flow for chapter 2 – Introduction.
Machine learning or prediction models section:
The author provided basic information of machine learning, but it did not link the physiochemistry properties prediction and motivations of using machine learning model. Although I understand that table 8 provided a lot of information for the current available papers using machine learning or models to predict physiochemistry properties, I would suggest that when the authors discuss about the machine learning techniques, they could also provide the examples of existing models using those techniques for prediction, which might have disadvantages or advantages. The authors can discuss about the reason or suggestion why it would better to use that machine learning techniques.
- We decided to reverse the subchapters to better reflect the sequence an analyst should follow when building a prediction model. It begins by emphasising the retention data and structural features, then moves on to unsupervised learning, and concludes with supervised learning. We have also relocated specific examples of those techniques to better show the connection between this and the previous chapter – Chapter 3.
- The discussion on ML techniques has been revised – Chapter 4.3.
Discussion section:
While Table 8 offers useful information, it is challenging for readers to grasp the messages the authors intend to convey to the community. The comments are quite general, and the second column contains excessive text, which may not be appropriate. I suggest making this section more concise and precise.
- Following the expansions of previous chapters, we decided that previous Table 8 was unnecessary. Additionally, it hindered the smooth introduction of some examples to avoid repetition. We have combined discussion and future perspectives to showcase the current state of the art and future challenges – Chapter 4.
Specific comments:
Table 1: In the second column, it showed ellipsis, leading to a lot of confusion for readers. I would suggest to make the table into paragraph or make the second column only showing some key words, and then discuss that in the manuscript.
- We have relocated Table 1 to the introduction, removed the ellipsis, and converted it into keywords – Table 1.
In line 171, please provide more discussion about ChromLogD.
- We have included some clarification about ChromLogD, but we decided not to initiate a discussion on it since later chapters only implement CHI – lines 60-72.
In lines 215-217, the authors only mentioned about Caco-2 and PAMPA. It would better to also mention MDCK, and the difference between these assays. The authors also mentioned that correlation is weak for compounds. Could you provide more detailed discussion about that? Because that would be very insightful information which the community would like to know.
- We have introduced the MDCK assay alongside Caco-2 and PAMPA, and describe their differences – lines 183-189.
- We have discussed the correlation of IAM membrane permeability with Caco-2 and MDCK membrane permeability, based on existing literature. We have highlighted key aspects of this correlation and emphasise differences related to the type of diffusion – lines 190-199.
Table 1 and 7 contain ellipsis, which is very difficult to follow.
- We have removed the ellipsis from Table 1 – Table 1.
- We have removed the ellipsis from Table 7 to Table 3.
I hope this new version will be more enjoyable to read and make a more significant contribution to its field.
Regards,
Damian Tuz

Reviewer 3 Report
Comments and Suggestions for Authors
My comments are included in the attachment.

Author Response
Dear Reviewer,
We want to thank you for taking the time to review our paper. We greatly appreciate your critics regarding our paper.
Before we address your comments line by line, we would like to point out that the manuscript has undergone significant revisions. The first reason for this was to incorporate the content of Table 8 into the core of the manuscript. The second reason was to enhance the depth and completeness of my work. Major changes include:
- Introduction: We relocated lipophilicity from the physicochemical section here to avoid repetition. This allowed us to establish a clear structure: physicochemical problem, golden standard, biomimetic chromatography solution.
- Chapter 2: We have combined biomimetic chromatography and physicochemical sections into one to strengthen their connection. This section also follows the structure outlined in the introduction. Additionally, we have expanded on two techniques that previously received only superficial treatment.
- Chapter 3: We decided to rearrange the subchapters to reflect better the sequence that an analyst should follow when building a prediction model. It begins by emphasising the retention data and structural features, then implements unsupervised learning, and concludes with supervised learning. We have also moved specific examples of these techniques to demonstrate better the connection between this chapter and the previous one.
- Chapter 4: We have integrated discussion and future perspectives to showcase the current state of the art and future challenges.
Regarding specific comments:
Major:
In my opinion, Table 8 should be more concise and compact. A review should be reader-friendly and include the most important information. I hope the authors can summarize this table and put some of the more detailed information in the Supporting Materials.
- After the expansions of earlier chapters, we decided that the previous Table 8 was unnecessary. Furthermore, it was hindering the smooth introduction of some examples to avoid repetition.
Minor:
Even though this paper is understandable, it contains many grammatical errors.
- We have carefully reviewed the paper for any grammatical errors. We hope this improvement is sufficient.
I hope this new version will be more enjoyable to read and make a more significant contribution to its field.
Regards,
Damian Tuz

Round 2
Reviewer 2 Report
Comments and Suggestions for Authors
- The revised manuscript offers a more detailed examination of how high-throughput biomimetic chromatography, combined with machine learning techniques, can predict potential drug properties. I appreciate that the authors revised the manuscript by implementing my suggestions.
- The manuscript is logically structured, but three main areas require improvement:
- Section 2.2 includes BBB permeability; however, Section 2.3 focuses exclusively on this topic. While acknowledging the importance of BBB permeability for CNS drug discovery, separating these sections may disrupt coherence. For instance, I would like to know the differences between the columns in IAM when testing normal cell permeability versus BBB permeability.
- Line 505-519: It would be beneficial to mention data splitting techniques, such as random splitting and scaffold splitting.
- Section 3.1 is titled “Molecular Descriptors vs. Fingerprints,” but fingerprints are a subset of the broader category of molecular representations, which includes 1D (string, fingerprint), 2D (graph), and 3D (structures). It would enhance clarity to either make this distinction clearer or include a discussion on molecular representations.
- Specific comments:
- Line 212: To maintain consistency with the other subsections, please replace “Mechanism” with “Technical Details.”
- Lines 231 and 277: For precision, please replace “skin” with “epithelium.”
- Line 309: Please specify the types of membrane receptors being referred to.
- Line 350: Ensure consistency in capitalization for all terms; use lowercase for "logD" and "ChromlogD," and uppercase for "LogP."
Author Response
Dear Reviewer,
We are pleased to observe that the initial revision has garnered a more favourable response. We appreciate the additional comments provided. Please find the detailed responses outlined below.
Regarding specific comments:
Section 2.2 includes BBB permeability; however, Section 2.3 focuses exclusively on this topic. While acknowledging the importance of BBB permeability for CNS drug discovery, separating these sections may disrupt coherence. For instance, I would like to know the differences between the columns in IAM when testing normal cell permeability versus BBB permeability.
- We have incorporated section 2.3 into section 2.2. Additionally, we emphasise the principal difference in methodology concerning permeability measurements that we identified, mainly the pH of the buffer and the types of molecular description – lines 220-225 and 243-245.
Line 505-519: It would be beneficial to mention data splitting techniques, such as random splitting and scaffold splitting.
- We have supplied further details concerning data partitioning methodologies, differentiating between random partitioning and one of the rational splitting techniques, namely scaffold splitting, which is especially pertinent in this context – lines 347-352.
Section 3.1 is titled “Molecular Descriptors vs. Fingerprints” but fingerprints are a subset of the broader category of molecular representations, which includes 1D (string, fingerprint), 2D (graph), and 3D (structures). It would enhance clarity to either make this distinction clearer or include a discussion on molecular representations.
- The initial paragraph of this subsection has been relocated to the preceding subsection to improve overall coherence. The subsection title has been amended to 'Molecular Representations,' and the initial paragraph has been dedicated to providing a clearer contextual placement of Molecular Descriptors and Molecular Fingerprints within the broader framework. Additionally, we have removed the potentially misleading content from the Molecular Fingerprints paragraph – Section 3.1.
Line 212: To maintain consistency with the other subsections, please replace “Mechanism” with “Technical Details”.
- We have updated 'Mechanism' to 'Technical Details' – line 162.
Lines 231 and 277: For precision, please replace “skin” with “epithelium”.
- We have replaced “skin” with “epithelium” – lines 180 and 237.
Line 309: Please specify the types of membrane receptors being referred to.
- We revised the previous statements to provide examples of various potential receptors – lines 252-258.
Line 350: Ensure consistency in capitalisation for all terms; use lowercase for “logD” and “ChromlogD” and uppercase for “LogP”.
- We ensured that log D in the appropriate context is presented in lowercase, while Log P is denoted in uppercase – lines 46, 49, 51, 53, 56, 66-67, 69, 258, 497, 549.
We hope this revision satisfactorily addresses all your comments.
Regards,
Damian Tuz

Reviewer 3 Report
Comments and Suggestions for Authors
The authors have carefully addressed my comments and the quality of the paper has been substantially improved. The paper can now go to publication.
Author Response
Dear Reviewer,
We are delighted to receive your positive feedback and recommendation for publication. We sincerely thank you for your time and constructive comments, which have helped us significantly improve the manuscript.
Regards,
Damian Tuz
